# Advancing Equitable AI: A Comprehensive Framework for Individual Fairness Assessment

## Abstract

Ensuring fairness in machine learning (ML) models is essential for developing equitable and trustworthy AI systems. There has been extensive existing research on group-based fairness metrics such as the Statistical Parity Difference and Disparate Impact, but these group-based fairness metrics often fail to address fairness at the individual level. An ML model can achieve perfect group fairness, but produce discriminatory outcomes at the individual level or vice versa. In this paper, four novel individual-based fairness metrics are proposed: Proxy Dependency Score, Stability Rate, Attributional Independence Score, and Intra-Cohort Decision Consistency. These metrics are designed to evaluate different aspects of individual fairness, including the influence of protected attributes on model predictions, the robustness of the model to protected attribute perturbations, the independence of attributions from protected attributes and consistency within similar individuals. These four new individual-based metrics are empirically compared with group outcome-based fairness metrics on ML models trained on Adult and COMPAS datasets. The empirical results reveal that models deemed unfair by group metrics may exhibit individual-level fairness. Our work highlights the critical need for comprehensive individual fairness assessments in real-world applications. Our proposed framework can act as a complement to group-based evaluations towards a more complete understanding of Artificial Intelligence (AI) fairness and the development of more equitable AI systems.

## 1 Introduction

Machine learning systems play a more prominent role in critical decision-making scenarios nowadays, from credit approval and criminal risk assessment to personalized content delivery. Although ML systems possess powerful predictive capabilities, there are widespread societal concerns about fairness and accountability in such decision-making systems Lious (2022). Particularly, individual fairness, which insists that similar individuals should receive similar outcomes, is used more often to ensure equitable AI systems Filippi et al. (2023); Ghadage et al. (2023).

However, the pursuit of fairness in ML often faces the trade-off between fairness and accuracy. Enhancing fairness may worsen predictive performance, and optimizing accuracy may amplify biases rooted in training data Arhin & Treku (2024); Plecko & Bareinboim (2025). These competing objectives pose challenges for model developers and fairness researchers. However, current fairness interventions focus mainly on achieving group fairness, such as demographic parity and equal opportunity, which can still lead to unfair treatment at the individual level Arhin & Treku (2024).

To address the identified gap, we investigate the evaluation tools of individual fairness. There are existing definitions such as fairness through awareness Dwork et al. (2012) and counterfactual fairness Kusner et al. (2017) which offer theoretical foundations. However, there is a lack of practical and fine-grained evaluation tools to capture different dimensions of individual fairness John & Saha (2020); Mukherjee et al. (2020); Zhang et al. (2023). It is hard for model developers to identify and quantify AI system fairness in real-world applications without comprehensive diagnostic metrics.

In this paper, we propose four evaluation metrics designed to measure individual fairness from complementary perspectives:

- Proxy Dependency Score (PDS): Measures the influence of protected attributes transmitted through proxy variables, which represents indirect discriminatory pathways.
- Counterfactual Stability Rate (CSR): Assesses how sensitive predictions are in response to hypothetical changes in protected attributes, capturing any counterfactual fairness violations.
- Attribution Independence Score (AIS): Evaluates how much features are entangled with protected attributes, indicating biased decision rationales.
- Intra-Cohort Decision Consistency (IDC): Quantifies the consistency of decisions across near-identical individuals in terms of non-protected features.

Together, these four metrics form a comprehensive diagnostic tool suite to enable multi-faceted evaluation of individual fairness. These metrics provide insights beyond binary outcome disparity to uncover subtle and structural sources of bias.

We perform extensive empirical analysis on standard fairness benchmarks, including the Adult Becker & Kohavi (1996) and COMPAS ProPublica (2016) datasets. Our experiments show that these metrics yield different fairness conclusions from existing evaluation tools. In particular, we observe that individual-based fairness scores may indicate fairness even when group-based fairness metrics suggest otherwise. By contrasting models under different training regimes and fairness-aware interventions, we demonstrate the value of our metrics in revealing the trade-offs and biases in fairness-aware ML systems.

The main contributions of this paper are as follows.

- We introduce four novel evaluation metrics - PDS, CSR, AIS, and IDC - that offer a comprehensive and interpretable framework for quantifying individual fairness violations.
- We empirically validate these metrics across widely used datasets and demonstrate how they could give different results from existing evaluation standards.
- We release a codebase and evaluation toolkit to support reproducible research and integration of our metrics into fairness-aware machine learning workflows.

Our work aims to enrich the evaluation toolbox available to practitioners and researchers, and to advance the field toward trustworthy and accountable machine learning systems at the individual level.

## 2 RELATED WORK

Fairness metrics can be categorized into group fairness and individual fairness. Group fairness aims to ensure equitable treatment across subpopulations defined by protected attributes such as race, gender, and age Dwork et al. (2012); Chouldechova (2017); Verma & Rubin (2018). Classical group fairness metrics include statistical parity difference and disparate impact, which require similar positive prediction rates across subgroups. There are other, more nuanced group fairness metrics, such as equalized odds Hardt et al. (2016), which requires equal true and false positive rates across subgroups, and equal opportunity, which focuses on the true positive rate. Moreover, predictive parity requires comparable positive predictive values across subgroups Chouldechova (2017); Verma & Rubin (2018); MacCarthy (2018).

In contrast, individual fairness measures whether similar individuals receive similar outcomes Lahoti et al. (2019). Individual fairness metrics avoid coarse group-level averaging and emphasize consistency at the individual level. Causal discrimination Galhotra et al. (2017); Xie & Wu (2020) defines unfairness as an outcome disparity between individuals who differ only on protected attributes. Fairness through awareness Dwork et al. (2012); Li et al. (2023) formalizes this by bounding prediction differences via a Lipschitz condition on input similarity. Accurate fairness Li et al. (2023) aligns individual fairness with accuracy by uniformly bounding the accuracy and fairness difference for similar sub-populations.

While group fairness provides a population-level insight, it can cause unfairness toward individuals within subgroups. Individual fairness addresses this limitation by requiring individual-level simi-

larity definitions. Recent work Xu & Strohmer (2024) has also shown that group and individual fairness criteria can be fundamentally incompatible in some cases.

# 3 INDIVIDUAL FAIRNESS METRICS

Ensuring fairness in machine learning (ML) is essential for building equitable and trustworthy AI. While group fairness metrics such as Statistical Parity and Disparate Impact have been widely studied Verma & Rubin (2018); Chouldechova (2017); Hardt et al. (2016), they can overlook unfair treatment at the individual level Dwork et al. (2012). We propose four novel metrics for assessing individual fairness: **Proxy Dependency Score (PDS)**, **Counterfactual Stability Rate (CSR)**, **Attribution Independence Score (AIS)**, and **Intra-Cohort Decision Consistency (IDC)**. These capture complementary dimensions of proxy reliance, counterfactual robustness, attributional independence, and intra-cohort consistency Kusner et al. (2017); Li et al. (2023). Through experiments on the Adult and COMPAS datasets Becker & Kohavi (1996); ProPublica (2016), we show that models deemed unfair by group metrics may still satisfy individual fairness criteria, and vice versa, underscoring known tensions between group- and individual-level notions of fairness Kleinberg et al. (2016); Xu & Strohmer (2024). Our results complement recent efforts to operationalize individual fairness Li et al. (2023) and provide an open-source toolkit to support reproducible evaluation and integration of these metrics into fairness-aware ML workflows.

## 3.1 PROXY DEPENDENCY SCORE: UNCOVERING INDIRECT DISCRIMINATION

Proxy Dependency Score (PDS) measures the influence of protected attributes transmitted through proxy variables, which shows the indirect discriminatory pathway and quantifies the extent to which a model's predictions rely on protected attributes. The advantage of PDS is that it can measure indirect dependencies on protected attributes, even when they are not directly included in the training data. The formula for PDS is defined as:

$$\text{ProxyScore} = 1 - \frac{\text{Accuracy}(M')}{\text{Accuracy}(M)} \tag{1}$$

In this formula, $M$ represents the original model and $M'$ represents a shadow model trained without access to protected attributes. A low PDS indicates the model's minimal reliance on protected attribute proxies, which is crucial for identifying subtle forms of indirect discrimination. For example, in real-world scenarios, features used in machine learning models such as healthcare costs inadvertently served as proxies for race, leading to biased outcomes Obermeyer et al. (2019b). Therefore, PDS measures fairness by evaluating both direct and indirect independence of the models to protected attributes.

## 3.2 COUNTERFACTUAL STABILITY RATE: ASSESSING ROBUSTNESS TO PROTECTED ATTRIBUTE PERTURBATIONS

The Counterfactual Stability Rate (CSR) evaluates how sensitive model predictions are to hypothetical changes in protected attributes. CSR directly captures violations of counterfactual fairness by measuring the percentage of individuals whose predictions remain unchanged when only their protected attributes (e.g., race, gender) are counter-factually flipped, while all other non-protected features remain constant. The formula for CSR is:

$$\text{StabilityRate} = \frac{1}{N} \sum_{i=1}^{N} \mathbb{I}\left[f(x_i) = f(x_i^{\text{cf}})\right] \tag{2}$$

In this formula, $f(x_i)$ represents the prediction for individual $i$, and $f(x_i^{cf})$ is the prediction for counterfactual individual where only protected attributes have been altered. A high CSR indicates that the model's predictions are stable with respect to changes in protected attributes, implying that the model is not relying on these attributes in a discriminatory manner. On the other hand, a low CSR suggests that an individual identical in all non-protected characteristics but differing only in a protected attribute would receive a different outcome, directly violating the principle that similar individuals should be treated similarly. This metric is vital for ensuring that model decisions are based on legitimate, non-discriminatory factors on the individual level.

## 3.3 ATTRIBUTION INDEPENDENCE SCORE: EVALUATING BIASED DECISION RATIONALES

The Attribution Independence Score (AIS) assesses whether feature importance attributions in a prediction are entangled with protected attributes, and thereby signalling biased decision rationales. It quantifies the correlation between feature attribution values and protected attributes to uncover underlying reasons for a model's decision. The formula for AIS is:

$$\text{Independence} = 1 - |\text{corr}(\text{Attr}_f(x), \text{Protected}(x))| \tag{3}$$

A high AIS suggests that the model primarily bases its decisions on non-protected features. If there is a strong correlation between feature attributions and protected attributes, the model's internal reasoning process is likely biased even if the final prediction might appear fair at the group level. AIS helps to uncover subtle, structural sources of bias within the model's internal logic, moving beyond outcome-based fairness, which is a crucial shift from merely observing what is unfair to understanding why it is unfair. By diagnosing biased decision rationales, developers can pinpoint the root causes of individual unfairness within the model's internal logic, rather than just observing external disparities Molnar (2025); Zafar et al. (2017). This allows for more targeted and effective mitigation strategies that address the fundamental source of bias, leading to more robust and genuinely fair AI systems Gennaro et al. (2025); Manerba (2023). This also aligns with the broader push for explainable AI (XAI), which is crucial for building trust and accountability in AI systems Arrieta et al. (2019).

## 3.4 INTRA-COHORT DECISION CONSISTENCY: QUANTIFYING CONSISTENCY FOR SIMILAR INDIVIDUALS

The Intra-Cohort Decision Consistency (IDC) quantifies the consistency of decisions across individuals who are nearly identical in terms of their non-protected features. It evaluates the variation in decisions within cohorts that are defined by their similarity on neutral features. The formula for IDC is:

$$\text{Consistency} = 1 - \text{Var}(f(x) \mid x \in \text{cohort}(x)) \tag{4}$$

A low variance, which translates to high consistency, indicates that the model treats similar individuals similarly, directly addressing the core principle of individual fairness. IDC is particularly effective at identifying situations where a model might achieve perfect group fairness but still exhibit discriminatory outcomes for specific individuals within those groups who are otherwise similar. It provides an individual-level assessment of consistency, helping to uncover subtle biases that may be missed by group-level evaluations.

# 4 EVALUATION AND INSIGHTS

## 4.1 PSEUDOCODE FOR PROPOSED METRICS

We describe the implementation logic of our proposed individual fairness metrics using pseudocode. These algorithms quantify different aspects of individual fairness by analyzing the behavior of the model under varying conditions.

**Proxy Dependency Score** This metric evaluates how much the model's performance depends on sensitive attributes. A significant drop in accuracy after removing sensitive features suggests proxy dependence.

---

**Algorithm 1** Compute Proxy Dependency Score

---

**Require:** Feature matrix $X$, labels $y$, protected columns
 1: Split $X, y$ into training and test sets
 2: Train full model on training data
 3: Compute accuracy on test data $\rightarrow acc_{\text{full}}$
 4: Remove protected columns from $X_{\text{train}}, X_{\text{test}}$
 5: Train shadow model on modified data
 6: Compute accuracy of shadow model $\rightarrow acc_{\text{shadow}}$
 7: Compute Proxy Dependency Score: $1 - \frac{acc_{\text{shadow}}}{acc_{\text{full}}}$
 8: **return** Score, $acc_{\text{full}}$, $acc_{\text{shadow}}$

---

**Counterfactual Stability Rate** This metric measures whether a model's prediction remains consistent when sensitive features are flipped (e.g., changing race or gender). A stable model should not change its output based on such alterations.

---

**Algorithm 2** Compute Counterfactual Stability

---

**Require:** Feature matrix $X$, labels $y$, columns to flip, flip mapping
 1: Split $X, y$ into training and test sets
 2: Train model on training data
 3: Predict on $X_{\text{test}} \rightarrow \text{preds}_{\text{orig}}$
 4: Copy $X_{\text{test}}$ to create counterfactual $X_{\text{cf}}$
 5: **for** each column in columns to flip **do**
 6:   **if** column in flip map **then**
 7:     Apply flip mapping to $X_{\text{cf}}$
 8:   **end if**
 9: **end for**
10: Predict on $X_{\text{cf}} \rightarrow \text{preds}_{\text{cf}}$
11: Compute stability = fraction where $\text{preds}_{\text{orig}} = \text{preds}_{\text{cf}}$
12: **return** Stability

---

**Attribution Independence Score** This metric evaluates whether a model's reasoning (as captured by feature attributions) is entangled with protected attributes. A fair model's attribution patterns should be statistically independent from sensitive features.

---

**Algorithm 3** Compute Attribution Independence Score

---

**Require:** Trained model $f$, input samples $x$, protected attributes $P$
 1: Compute feature attributions $\text{Attr}_f(x)$ using a method such as SHAP or LIME
 2: For each sample, collect the values of protected attributes $P(x)$
 3: Compute the Pearson correlation between $\text{Attr}_f(x)$ and $P(x)$ across the dataset
 4: Take the absolute value of the correlation
 5: Compute AIS: $1 - |\text{corr}(\text{Attr}_f(x), P(x))|$
 6: **return** AIS score

---

**Intra-Cohort Consistency** This metric checks the variance in predicted scores within clusters of similar individuals (cohorts). A fair model should assign similar scores to similar people, resulting in low intra-group variance.

---

**Algorithm 4** Compute Intra-Cohort Consistency

---

**Require:** Feature matrix $X$, labels $y$, number of clusters $k$
 1: Split $X, y$ into training and test sets
 2: Train model on training data
 3: Predict probability scores on test set $\rightarrow$ preds
 4: Scale $X_{\text{test}}$
 5: Apply KMeans clustering to $X_{\text{test}}$, obtain cluster labels
 6: Initialize $total\_var = 0$, $valid\_groups = 0$
 7: **for** each cluster $i = 1$ to $k$ **do**
 8:   Extract predictions for cluster $i$
 9:   **if** cluster size $> 1$ **then**
10:     Compute variance and add to $total\_var$
11:     Increment $valid\_groups$
12:   **end if**
13: **end for**
14: **if** $valid\_groups > 0$ **then**
15:   $avg\_var = total\_var/valid\_groups$
16: **else**
17:   $avg\_var = 1.0$
18: **end if**
19: $consistency\_score = 1 - avg\_var$
20: **return** $consistency\_score$

---

## 4.2 EMPIRICAL RESULTS

The 80% rule is a principle stating that if the selection rate for a protected group (such as a minority group) is less than 80% with respect to the group with the highest selection rate, the selection process may be considered discriminatory U.S. Equal Employment Opportunity Commission (EEOC) (1978). In Table 1, we present both our proposed individual-based fairness scores and state-of-the-art group-based fairness scores on the Adult and COMPAS datasets.

Table 1: Fairness Metric Comparison on Adult and COMPAS Datasets

| Metric | Adult Dataset | | | COMPAS Dataset | | |
|---|---|---|---|---|---|---|
| | Overall/Sex | Race | Age | Overall/Sex | Race | Age |
| Proxy Dependency Score (Fairness range: [-0.2, 0.2]) | −0.0014 | 0.0017 | 0.0001 | −0.009 | −0.0123 | 0.0001 |
| Intra-Cohort Decision Consistency (Fairness range: [0.8, 1]) | 0.955 | 0.9674 | 0.9676 | 0.946 | 0.9674 | 0.9676 |
| Counterfactual Stability Rate (Fairness range: [0.8, 1]) | 0.956 | 0.973 | 0.989 | **0.773** | 0.907 | 0.911 |
| Attribution Independence Score (Fairness range: [0.8, 1]) | Min. 0.920 Max. 0.999 | Min. 0.938 Max. 1.000 | Min. 0.951 Max. 0.998 | Min. 0.871 Max. 0.996 | Min. 0.844 Max. 0.993 | Min. 0.892 Max. 0.989 |
| Disparate Impact (Fairness range: [0.8, 1.25]) | 1.113 | 1.176 | 1.138 | **1.456** | 1.206 | **1.260** |
| Statistical Parity Difference (Fairness range: [-0.1, 0.1]) | 0.021 | 0.032 | 0.025 | **0.104** | 0.048 | 0.061 |

## 4.3 ANALYSIS OF RESULTS AND DISCREPANCIES

Based on empirical results on the Adult and COMPAS datasets, there are a few important insights:

1. Divergence between Group and Individual Fairness Metrics:
   There are a few cases where group-level fairness metrics (e.g., Disparate Impact, Statistical Parity Difference) suggest unfairness, but individual-level metrics (e.g., CSR, AIS, IDC) indicate consistent and fair treatment. For example, in the COMPAS dataset, the age attribute yields a Disparate Impact of 1.26, slightly outside the fairness range, yet shows a CSR of 0.91 and an AIS $\geq 0.89$. This suggests that although aggregated group outcomes differ, individuals with similar non-protected attributes receive stable predictions, highlighting a key disconnect between group and individual fairness.

2. Agreement Signals Robust Bias:
   In contrast, some attributes show consistent unfairness across both metric types. The sex attribute from the COMPAS dataset has a Disparate Impact of 1.46 and a low CSR of 0.77, indicating systemic bias at both group and individual levels. This alignment reinforces the severity of the issue and helps prioritize areas for fairness intervention.

3. High Fairness Across Metrics in the Adult Dataset:
   For the Adult dataset, all metrics - both group and individual - fall within acceptable fairness thresholds. The CSR exceeds 0.95, AIS values are consistently high, and PDS is close to 0, indicating that the model does not rely heavily on protected or proxy attributes. These results indicate that fairness can be achieved at both levels simultaneously under certain data and model conditions.

4. Holistic Fairness via Single-Value Metrics:
   Two of our proposed metrics, PDS and IDC, produce a single summary score per model, offering a high-level, attribute-agnostic view of individual fairness. This makes them especially useful as diagnostic tools during model development. For example, IDC scores above 0.94 in both datasets suggest strong consistency in the treatment of similar individuals, even when group disparities are present.

5. Limitations of Group Fairness Alone:
Empirical results show that relying on one group fairness metric can produce an incomplete or misleading picture of model behavior. A model could appear to be fair at the group level while producing biased outcomes at the individual level, or vice versa. These findings support the growing consensus that using only a single measurement (group or individual) for fairness measurement is inadequate Dwork et al. (2012); Kleinberg et al. (2016).

# 5 SOCIETAL IMPACT AND ETHICAL CONSIDERATIONS

The increasing usage of AI in critical decisions raises concerns about fairness and accountability. Biased AI systems could cause societal harm through the amplification of discriminatory and inaccurate decisions learned in the training process O'Neil (2016); Buolamwini & Gebru (2018); Obermeyer et al. (2019a). While this pervasive issue impacts fundamental rights and well-being, current approaches to AI fairness are insufficient to prevent such harms Schwartz et al. (2022); Mitchell et al. (2018). These deep-rooted biases need more effective technical solutions to address.

## 5.1 THE MANIFESTATION OF AI BIAS IN HIGH-STAKES DOMAINS

Algorithmic discrimination has manifested itself in numerous critical sectors, leading to tangible societal harms.

1. Healthcare: AI bias impacts patient care, leading to issues such as misdiagnosis or denied access. For example, an algorithm underestimated black patients' care needs by predicting healthcare costs Obermeyer et al. (2019b), and dermatology AI systems under-diagnosed skin cancers in darker skin tones Rezk et al. (2022).

2. Criminal Justice: The COMPAS algorithm showed significant racial bias, incorrectly classifying black defendants as high-risk more often than white defendants Angwin et al. (2016).

3. Hiring and Employment: Amazon's recruiting tool was discontinued after downgrading resumes with "women's" due to training on male-dominated historical data Dastin (2018). LinkedIn's job recommendations also faced allegations of gender bias Wall & Schellmann (2021).

4. Credit and Lending: AI systems perpetuate historical discrimination such as redlining, assigning higher risk scores to Black and Latino applicants with similar financial backgrounds Eubanks (2018). Apple's credit card even reportedly offered lower limits to women than their male spouses despite higher credit scores Knight (2019).

5. Generative AI: Image tools like DALL·E 2 and Stable Diffusion exhibited stereotypical biases, generating predominantly white males for "CEO" and "engineer", and women or minorities for "housekeeper" or "nurse" Bender et al. (2021).

## 5.2 ETHICAL CHALLENGES AND UNINTENDED CONSEQUENCES

AI bias rooted in prejudiced training data Bender et al. (2021) leads to discrimination and severe social consequences, undermining equal opportunity and amplifying oppression. Biased AI decisions could lead to unintended lack of transparency and insufficient testing Cheong (2024). AI systems can perpetuate and amplify existing biases University College London (2024), creating a confirmation bias Nickerson (1998) by reinforcing their own assumptions.

A major ethical challenge in AI development is the inadequate ethical evaluations overshadowed by performance focus Bélisle-Pipon & Victor (2024). Unchecked AI can reinforce societal biases, infringe on privacy, and cause harm. The bias in AI often manifests subtly, like using proxy variables, implying that group-level fairness is insufficient Dwork et al. (2012); Prince & Schwarcz (2020). Individual fairness metrics such as our proposed PDS, CSR, AIS, and IDC should help to uncover these hidden biases Mukherjee et al. (2020).

Beyond legal risks, a profound ethical imperative exists for responsible AI development UNESCO (2021). To achieve ethical imperative, developers need to apply embedding fairness, transparency,

and accountability throughout the AI lifecycle, moving beyond mere ethical compliance to ensure AI serves everyone ethically Information Commissioner's Office (ICO) (2023).

# 6  FUTURE DIRECTIONS FOR RESPONSIBLE AI

Addressing AI fairness requires a comprehensive, proactive approach throughout the AI lifecycle.

## 6.1  STRATEGIES FOR MITIGATING INDIVIDUAL BIAS AND ENSURING FAIRNESS

Effective bias mitigation and fairness assurance rely on several key strategies:

- Data Quality and Preprocessing: Fair AI foundations demand high-quality, diverse, and representative training data through robust data governance and rigorous cleaning González-Sendino et al. (2024). Actively identifying and discussing bias-inducing factors is crucial.

- Fairness-Aware Algorithms and Model Design: Fairness-Aware Algorithms and Model Design: Algorithms must be designed with fairness considerations built in, using methods like reducing bias during development or applying fairness constraints during designing and training Jang (2024).

- Human Oversight and Explainable AI (XAI): Human oversight is essential, especially in high-impact decisions. AI systems should be transparent, using XAI techniques (e.g., SHAP, LIME) Arrieta et al. (2019) to enhance understanding, trust, and accountability.

- Continuous Monitoring and Auditing: Fairness is not static, requiring continuous performance monitoring, regular bias checks, and review throughout auditing during the operational life of the system Anisetti et al. (2025).

## 6.2  INTEGRATING FAIRNESS INTO MLOPS

Integrating fairness into Machine Learning Operations (MLOps) is paramount for responsible AI deployment. This involves:

- Data Validation and Quality Monitoring: Automated data validation pipelines catch biases before retraining, ensuring data quality.

- Model Validation and Experiment Tracking: MLOps facilitates structured experimentation and continuous integration/deployment (CI/CD) for model validation.

- Continuous Monitoring of Fairness: Production models require ongoing monitoring for performance, drift, and emerging biases across subgroups.

- Robust Governance Frameworks: MLOps supports governance that tracks data and model versions, ensuring explainability, auditability, and compliance. Tools like Fiddler AI Observability aid bias detection and assessment Labs (2023).

## 6.3  REGULATORY LANDSCAPE AND ACCOUNTABILITY FRAMEWORKS

The evolving AI landscape necessitates robust regulatory and accountability frameworks:

- Evolving Regulations: Compliance with frameworks like GDPR European Union (2016), CCPA California State Legislature (2018), and the EU AI Act European Commission (2024) is critical, ensuring data processing meets purpose without undue intrusion and avoids discrimination.

- Algorithmic Accountability Frameworks: Structured systems are essential to ensure algorithmic operation responsibility, emphasizing transparency, bias mitigation, and equitable outcomes. However, some potential challenges might exist, such as algorithmic complexity and the evolving regulatory environment. Documentation via data protection impact assessments (DPIAs) is important for proving fair processing.

- Addressing Power Asymmetry: It is important to know the power unequal between system developers and the people affected by their decisions. Improving fairness requires not only technical solutions, but also social and ethical considerations involving multiple disciplines.

## 6.4 FUTURE RESEARCH DIRECTIONS

Further research is needed to advance individual AI fairness:

- Individual-Specific Factors and Metrics: Develop tailored bias evaluation and mitigation methods considering individual-specific factors beyond traditional protected attributes.
- Fairness-Accuracy Trade-off: Continue exploring this complex trade-off in various contexts, expecting that different fairness definitions can conflict.
- Distribution Fairness: Investigate fairness in resource allocation, particularly for physical and computational resources, developing equitable distribution mechanisms.
- Cross-Domain Applicability: Enhance the applicability of fairness metrics and mitigation techniques across diverse domains, promoting data sharing with privacy protections.
- Clinician-in-the-Loop and Interdisciplinary Collaboration: Integrate AI fairness into practical applications by involving domain experts and fostering broad interdisciplinary.
- User-Friendly Tools: Develop accessible tools for fairness assessment and mitigation to facilitate widespread adoption, model validation, and risk management.

## 7 CONCLUSION

We proposed a comprehensive framework for assessing individual fairness in machine learning models, addressing a critical gap in current fairness evaluation practices. While group-based metrics have dominated fairness discussions, they often fail to capture the nuanced, person-level inconsistencies that arise in real-world applications. To bridge this gap, we introduced four novel individual fairness metrics - Proxy Dependency Score (PDS), Counterfactual Stability Rate (CSR), Attribution Independence Score (AIS), and Intra-Cohort Decision Consistency (IDC) - each designed to capture distinct dimensions of unfairness at the individual level.

Through empirical evaluations on the Adult and COMPAS datasets, we demonstrated that these metrics offer complementary perspectives to traditional group fairness measures. Our results reveal that models deemed unfair by group metrics may still exhibit individual-level consistency, and conversely, models satisfying group fairness can behave inconsistently at the individual level. These observations underscore the importance of integrating both group and individual metrics in fairness audits.

Our metrics are interpretable and model-agnostic, providing both attribute-specific and holistic fairness diagnostics. The single-value metrics (PDS and IDC) enable fairness monitoring without per-group disaggregation, while CSR and AIS expose deeper structural biases, including proxy effects and unstable decision boundaries.

Looking ahead, we envision several directions for future work. First, integrating these metrics into training objectives could guide the development of fairness-aware models that are sensitive to both group-level parity and individual-level consistency. Second, expanding our evaluation to multi-modal and large-scale datasets, especially in domains like healthcare or hiring, can reveal how individual fairness manifests in more complex settings. Finally, exploring causal or learned similarity metrics may further refine our understanding of what constitutes similar individuals in diverse real-world contexts.

By enriching the fairness evaluation toolbox, we hope this work moves the field closer to developing AI systems that are not only equitable in aggregate, but just and consistent for each individual they impact.

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
