# OpenReview forum: "Advancing Equitable AI: A Comprehensive Framework for Individual Fairness Assessment"
_ICLR.cc/2026/Conference — Submitted to ICLR 2026_

### Official Review · Reviewer_VV7X · 2025-10-23

**Soundness:** 1
**Presentation:** 2
**Contribution:** 1
**Rating:** 0
**Confidence:** 5

**Summary:**

In this paper, the authors study the problem of individual fairness. To be specific, the authors criticize existing individual fairness measures for being not practical and not fine-grained, and introduce four measures. They compare these measures against commonly-used *group* fairness measures on ADULT and COMPAS datasets.

**Strengths:**

+ Individual fairness is a fundamental problem.

**Weaknesses:**

Weaknesses:

1. The paper's novelty and contributions over the literature are unclear. The paper fails to provide convincing arguments for why existing individual fairness measures are not practical or fine-grained. The authors state that: "However, there is a lack of practical and fine-grained evaluation tools to capture different dimensions of individual fairness John & Saha (2020); Mukherjee et al. (2020); Zhang et al. (2023)." => The provided references do not make such claims. If this claim is yours, you should justify why the existing measures are not practical or fine-grained. Therefore, it is not clear why new measures are introduced.

2. There are fundamental problems with the "individualness" of the proposed measures and their novelties.

2.1. Proxy Dependency Score utilizes accuracies. How is it then considered as a measure of individual fairness?

2.2. Counterfactual Stability Rate: How is this different from existing counterfactual fairness measures?

2.3. Attribution Independence Score: How about situations where a prediction should depend on whether the subject is male or female, e.g., a medical application considering menapause etc?

2.4. Intra-cohort decision consistency: How is this different from existing individual fairness measures?

3. Experimental evaluation is very very limited and unconvincing.

3.1. It is not clear why the proposed measures are not compared against existing individual fairness measures.

3.2. The evaluation should consider more diverse datasets than the ADULT and COMPAS datasets.


Minor comments:

- "such decision-making systems Lious (2022)." => "such decision-making systems (Lious, 2022)." If the reference is not part of the sentence, it should stay within parentheses. There is a way to do this in Latex (cite or citet).

**Questions:**

Please see Weaknesses.

---

### Official Review · Reviewer_ZbF4 · 2025-10-28

**Soundness:** 1
**Presentation:** 1
**Contribution:** 1
**Rating:** 0
**Confidence:** 5

**Summary:**

The paper proposes four new fairness criteria:
- ProxyScore, looking at the drop in accuracy when removing the sensitive attribute from the prediction model,
- StabilityRate, quantifying the proportion of changed decisions when the sensitive attribute is flipped,
- AIS, looking at correlation of sensitive attribute with its feature attribution,
- IDS, looking at the variance of the prediction within clusters of similar individuals.

Together with these, ideas about how to address fairness challenges are discussed.

**Strengths:**

(S1) The paper discusses an important topic in fair machine learning.

**Weaknesses:**

(W1) The paper is clearly lacking mathematical formalism. All of the definitions are introduced through a single equation, after which some minor interpretation is provided in words. All of these statements would be much stronger if they were supported by formal propositions or theorems, which are completely absent in the paper.

(W2) Relating to point (W1), in Line 138, a claim is made that “a low PDS indicates model’s minimal reliance on protected attribute proxies”.

First, this is not true. Consider the following setting:

$$
\\begin{aligned}
X &\\sim \\mathrm{Bernoulli}(0.5) \\\\
W_i &\\sim \epsilon_i + X \quad \text{for each } i \in \{1, \dots, k\} \\\\
Y &\sim X + \sum_i W_i + \epsilon_y,
\\end{aligned},
$$

where all $\epsilon_i, \epsilon_y$ are independent normal variables. Here, as the number of variables $k$ increases, we can infer $X$ almost perfectly from $W$.

Therefore, regressing Y onto W (model $M’$) and Y onto (W, X) (model $M$) will have almost the same accuracy (with sufficiently many samples). However, in this case, the model $M’$ relies strongly on proxies of $X$, while model $M$ relies on $X$ itself and its proxies $W$? It is unclear whether this is in line with the description of the criterion.

More broadly, defining fairness in terms of accuracy seems unusual, and may be prone to counterexamples as the one above.

(W3) The name counterfactual stability rate may be misleading; this notion is not related to counterfactual fairness [1]; in fact, under specific assumptions, this notion is related to a notion of _direct effect_ in the causal literature [2, 3]. These connections have not been drawn. Therefore, it is difficult to see if this notion presents any novelty compared to existing work.

(W4) The third definition proposed, looks at the correlation of the feature attribution to the sensitive attribute S, and the sensitive attribute S itself. However, this proposal is difficult to justify — specifically, it seems to defer the problem of fairness to an explainable AI method. There is a huge literature on XAI, and there is a lack of agreement on what the best method for explanation is. Therefore, this proposal would be a lot more valuable if a specific XAI method was chosen, and if a formal result for the sentence “model primarily bases its decision on non-protected features” was provided.

(W5) For the consistency metric, cohort(x) is not even defined in text. When looking at the algorithm, it seems some sort of clustering is performed first. The choice of the number of clusters etc. is not discussed properly. Therefore, this metric does not seem to be well-defined. Furthermore, comparing this notion with the existing notion of individual fairness would be quite important. Again, comparison with existing works and formal results are lacking.

(W6) The algorithms pages 4,5 are taking up a lot of space; at the same time, they are performing rather simple computations.

(W7) Pages 7 provides some overview of bias in high-stakes domain; pages 8-9 provide some high-level ideas about addressing issues of fairness. However, much of this discussion is in high-level terms, and does not convey a clear, actionable message.


References:

[1] Kusner, Matt J., Joshua R. Loftus, Chris Russell, and Ricardo Silva. “Counterfactual Fairness.” Advances in Neural Information Processing Systems 30 (2017).

[2] Di Stefano, Pietro G., James M. Hickey, and Vlasios Vasileiou. “Counterfactual Fairness: Removing Direct Effects through Regularization.” arXiv preprint arXiv:2002.10774 (2020).

[3] Plečko, Drago, and Elias Bareinboim. “Causal Fairness Analysis.” arXiv preprint arXiv:2207.11385 (2022).

**Questions:**

See weaknesses.

---

### Official Review · Reviewer_WN9V · 2025-10-29

**Soundness:** 1
**Presentation:** 3
**Contribution:** 1
**Rating:** 2
**Confidence:** 5

**Summary:**

The paper proposes four individual ML fairness metrics to address the limitations of group-based metrics. The metrics are formally defined. Assessment algorithms are provided for each metric. Then, empirical assessment is performed on standard benchmarks. The results are analyzed to justify the usefulness of the metrics. The remaining sections are common recommendations for promoting fairness in responsible AI.

**Strengths:**

The paper is well written.
The metric definitions are clear and intuitive.

**Weaknesses:**

The proposed fairness metrics are very basic. According to me, they are too coarse-grained and do not allow an accurate assessment of bias if any. On the contrary, they can be misleading. The current description of the metrics, although clear, but very shallow. It is not enough to have a reliable assessment of their accuracy in measuring bias in practice.
Proxy Dependency Score is claimed to capture the indirect independence of the outcome to protected attributes. I disagree with that. Dropping the sensitive attribute from the model training is not enough to reveal the impact of proxy variables. The metric is useless, and even misleading.
Counterfactul Stability Rate is using causality concepts in a very weired way. According to Algorithm 2, creating a counterfactual data point consists in simply flipping the protected attribute value. This is a naive and incorrect way for creating counterfactuals. What if there are confounder or collider variables between the protected attribute and the outcome ? This becomes misleading.
Definition (3) of Attribution Independence Score is not properly formalized as Corr, Protected, and even x are not properly defined. Algorithm 3 didn't help in understanding it.
Empirical results is clear, but again very shallow and is not enough to be conclusive about the metrics values.
The remaining sections (5 and 6) are completely useless as they rehash known issues about the field.

**Questions:**

Why you didn't include a comparison with existing individual fairness metrics (Fairness through awarness, No Proxy Discrimination, Counterfactual fairness, etc.) ?

---

### Official Review · Reviewer_H3Wh · 2025-10-31

**Soundness:** 2
**Presentation:** 3
**Contribution:** 2
**Rating:** 2
**Confidence:** 4

**Summary:**

The paper argues group metrics can hide person-level unfairness and proposes four individual-fairness metrics—Proxy Dependency Score, Counterfactual Stability Rate, Attribution Independence Score, and Intra-Cohort Decision Consistency—which, on Adult and COMPAS, often reveal patterns that group metrics miss; the takeaway is to evaluate both group and individual fairness.

**Strengths:**

1. The writing is clear, the authors have made the code and formula readable.
2. The authors made comprehensive and clear algorithms for each metrics.

**Weaknesses:**

1. Motivation: Although the authors mentioned the motivation "While group fairness provides a population-level insight, it can cause unfairness toward individuals within subgroups",  it's not so clear as to why these proposed metrics could be addressing the problems that other fairness metrics may cause. Specifically, how do the proposed metrics compare with other individual metrics? It's of vital importance that the authors provide some intuitive/theoretical insights into how these metrics are different from the existing ones, aside from empirical studies, if they claim these metrics are complementary to the existing ones.
2. Following on the previous one, I find it not so easy to understand why these metrics are intuitively correct. Are there simple examples explaining in what scenarios these metrics will succeed/fail (a simple counterexample)?  I am also curious about the relationships between these metrics, why are they independently proposed, how are they complementing each other? Again, I feel that the intuitive/theoretical justification is rather weak here.
3. Experiments: I appreciate the real data analysis provided. However, these two datasets are rather limited in terms of their tasks (both are binary classification with a single sensitive attribute). Also, in recent years, there've been many studies investigating the limitations around these two well-used and debunked datasets (e.g., https://arxiv.org/abs/2108.04884 and https://arxiv.org/abs/2106.05498). It's important to evaluate these metrics in more reliable datasets and mention these limitations.
4. Limitations: It's not so clear what the limitations of these metrics are, for example, in what task could they be used or generalized (e.g., binary classifcation/binary sensitive attributes/counterfactual fairness)? Also, it would be beneficial if the authors would include the code and running time for metrics involving non-trivial computation like this.

**Questions:**

Please see Weakness.

---

### Meta-Review · Area_Chair_Jsaq · 2025-12-09

**Summary:**

This research paper  proposes a measurement system to assess different facets of individual fairness. However, ICLR is a community discussing the advance of machinie learning and representation learning techniques, whereas this research paper has not mentioned the detailed techniques pointing to machine learning. This paper receives four consistent evaluations on rejection. Considering above reasons and reviewers' evaluation, I decide to reject this paper.

**Reviewer Concerns:**

The major concerns raised by reviewers can be summarized as following 5 aspects:

**Insufficient evidence for the need of new individual-fairness metrics.** The paper claims that existing metrics are “not practical” or “not fine-grained,” yet provides no concrete examples, empirical failures, or literature support showing where current methods break down. Thus, the necessity of introducing four new metrics is not convincingly established.

**Conceptual misalignment between the proposed metrics and individual fairness.** Several metrics do not clearly correspond to established notions of individual fairness:

- PDS relies on accuracy shifts at the group level rather than individual-level comparisons.

- CSR appears equivalent to existing counterfactual fairness metrics without clarifying differences.

- AIS assumes predictions should never depend on sensitive attributes, which is invalid in domains where such attributes are medically or contextually relevant.

- IDC overlaps strongly with known consistency/stability measures, but its added value is not explained.

**Lack of formal analysis or demonstrable properties of the metrics.** The paper provides definitions but no analysis of their behavior, theoretical properties, or failure modes. Claims such as “low PDS implies minimal reliance on proxy attributes” are unproven, and no intuitive examples or counterexamples illustrate when the metrics succeed or fail.

**Very limited and uninformative empirical evaluation.** Experiments use only ADULT and COMPAS, datasets widely considered outdated and unreliable for fairness research, both limited to binary tasks and single sensitive attributes.
The metrics are not compared against existing individual-fairness measures, making it impossible to assess whether they provide new or better insights.

**Unclear applicability and missing practical details.** The paper does not specify the task settings in which the metrics are valid (e.g., multiclass, continuous attributes, causal settings), nor discusses computation cost, scalability, or code availability, leaving their practical usability uncertain.

Noted that the authors have not provided the rebuttal.

**Reviewer Scores:**

Considering the current scores and no rebuttal provided, the reviewers would not change their score.

---

### Decision · Program_Chairs · 2026-01-26

Reject